# Overexpression of a *Fragaria vesca* 1R-MYB Transcription Factor Gene (*FvMYB114*) Increases Salt and Cold Tolerance in *Arabidopsis thaliana*

**DOI:** 10.3390/ijms24065261

**Published:** 2023-03-09

**Authors:** Wenhui Li, Peng Li, Huiyun Chen, Jiliang Zhong, Xiaoqi Liang, Yangfan Wei, Lihua Zhang, Haibo Wang, Deguo Han

**Affiliations:** 1Key Laboratory of Biology and Genetic Improvement of Horticultural Crops (Northeast Region), Ministry of Agriculture and Rural Affairs, National-Local Joint Engineering Research Center for Development and Utilization of Small Fruits in Cold Regions, College of Horticulture & Landscape Architecture, Northeast Agricultural University, Harbin 150030, China; 2Key Laboratory of Horticultural Crops Germplasm Resources Utilization, Ministry of Agriculture and Rural Affairs of the People’s Republic of China, Research Institute of Pomology, Chinese Academy of Agricultural Sciences, Xingcheng 125100, China; 3Institute of Agricultural Processing Research, Ningbo Academy of Agricultural Sciences, Ningbo 315040, China

**Keywords:** *Fragaria vesca*, FvMYB114, salt stress, cold stress, *Arabidopsis thaliana*

## Abstract

The MYB (v-MYB avian myeloblastosis viral oncogene homolog) transcription factor (TF) family has numerous members with complex and diverse functions, which play an indispensable role in regulating the response of plants to stress. In this study, a new 1R-MYB TF gene was obtained from *Fragaria vesca* (a diploid strawberry) by cloning technology and given a new name, *FvMYB114*. According to the subcellular localization results, FvMYB114 protein was a nuclear localization protein. Overexpression of *FvMYB114* greatly enhanced the adaptability and tolerance of *Arabidopsis thaliana* to salt and low temperature. Under salt and cold stress, the transgenic plants had greater proline and chlorophyll contents and higher activities of superoxide dismutase (SOD), peroxidase (POD) and catalase (CAT) than the wild-type (WT) and unloaded-line (UL) *A. thaliana*. However, malondialdehyde (MDA) was higher in the WT and UL lines. These results suggested that *FvMYB114* may be involved in regulating the response of *A. thaliana* to salt stress and cold stress. *FvMYB114* can also promote the expression of genes, such as the genes *AtSOS1/3*, *AtNHX1* and *AtLEA3* related to salt stress and the genes *AtCCA1*, *AtCOR4* and *AtCBF1/3* related to cold stress, further improving the tolerance of transgenic plants to salt and cold stress.

## 1. Introduction

When plants grow in the natural environment, due to the variability of environmental factors, they will inevitably suffer from various adverse environmental conditions, which will hinder their growth and development. All kinds of biotic and abiotic stresses pose a great threat to the normal life activities of plants, and extremely conditions can cause irreversible damage [1]. Beyond that, drought, low temperature, heavy metal pollution, high salt, nutrient deficiency and other abiotic stresses can cause significant harm to horticultural crops, which is a problem that many countries around the world are facing and urgently need to solve [2,3,4]. Improving the living conditions of plants under an adverse environment and improving the yield and quality of horticultural crops are pressing requirements. Increasing studies have proven that plants can perceive stimulus signals from the outside world and transmit them to cells to produce adaptive response mechanisms [5,6]. With the rapid development of molecular biology technology, gene manipulation has become an effective method to improve the adaptability of plants to the external environment. Among them, TFs and their regulatory functions play an important role in current plant research.

As one of the largest TF families in the plant genome, MYB, is widely involved in the response process to plant environmental factors [7]. Most MYB proteins play the role of TF. MYB TF is the term for a group of eukaryotic TFs with one or more conservative domain defined by 1–4 MYB repeats, which is composed of 51 or 52 amino acid (aa) residues. These aa residue pairs can form a DNA recognition helix that makes direct contact with DNA [8]. MYB TF is divided into four categories according to the number and arrangement characteristics of DNA-binding-domain MYB, namely 1R-MYB (MYB related), 2R-MYB (R2R3-MYB), 3R-MYB (R1R2R3-MYB) and 4R-MYB (R1R2R2R1/2-MYB) [9,10,11].

MYB TF was found in *Zea mays* first [12], and since, a variety of MYB gene families have been identified in many kinds of plants [13,14]. In addition, a large number of studies have proven that MYB TF can make adjustments to enable plants to cope with harmful environments when plants are subject to external interference. Bian et al. [15] found that under the condition of high salt and water shortage, the germination rate and seedling rate of seeds were significantly increased due to the overexpression of *GmMYB811*, which enhanced the ability of soybean to cope with salt stress and drought stress during seed germination. *FvMYB24* was shown to bind the *SOS1* promoter as a positive regulator that mediates salt tolerance in strawberry plants [16]. *MdMYB88* and *MdMYB124* can positively regulate the expression of cold tolerance and cold response genes in apple and *Arabidopsis* under cold stress through CBF-dependent and CBF-independent pathways [17]. Although there have been many previous studies on the function of MYB TF, they mainly focused on the regulation of volatile substances, such as anthocyanin and flavanol accumulation in strawberry fruits [18,19,20], while there have been few reports on the function of the MYB gene in salt and cold resistance of strawberry. The genetic transformation system of *F. vesca* is stable, meaning it can be used as the dominant subgenome of cultivated strawberry.

Among the pathways of plant response to various stresses, the role of the regulatory network of functional gene expression mediated by TFs cannot be ignored. MYB TF plays a key role in such signal networks. MYB can improve plant stress resistance by regulating the expression of multiple stress-related genes [21,22,23]. It was found that *XsMYB30* may directly or indirectly regulate the expression of defense-related genes, such as *XsPOD*, *XsSOD*, *XsCAT1*, *XsP5PCS*, *XsNCED1*, *XsABI3*, *XsABI5* and *XsABF2*, to endow the yellowhorn with drought and salt tolerance [24]. Jung et al. found that under drought and high salt conditions, the overexpression of *AtMYB44* in *Arabidopsis* could inhibit the expression of negative regulators (*HAB1/2*, *AtPP2CA*, *ABI1*) in the ABA signal pathway, thus enhancing the tolerance of *A. thaliana* to drought and salt, so it could grow normally in this vicious environment [25]. Moreover, MbMYBC1 can promote the expression of cold response genes such as *DREB1A*, *COR15a*, *ERD10B* and *COR47* under cold stress [26].

In this study, a new MYB gene, *FvMYB114*, was isolated and cloned from *F. vesca*, and its role in salt and cold tolerance was characterized in *A. thaliana*. All data indicated that *FvMYB114* may play an important role in improving the salt tolerance and cold tolerance of strawberry. The results provide a scientific basis for further studies of the function of *FvMYB114* in strawberry salt and cold tolerance. They also provide a theoretical basis for further research on the molecular biological function of the *MYB* TF gene in the strawberry stress-resistance process.

## 2. Results

### 2.1. Cloning and Bioinformatic Analysis of FvMYB114

Sequence analysis results are shown in Appendix A, indicating that *FvMYB114* had a full length of 300 bp (http://www.rosaceae.org/mRNA/11734785 (accessed on 25 December 2022); name: Fv2339_5g36520-mRNA-2 | GDR (rosaceae.org (accessed on 25 December 2022); http://bioinformatics.towson.edu/strawberry/ (accessed on 14 March 2022); gene ID: gene26513). The theoretical molecular weight of the protein encoded by *FvMYB114* was 11.626 kDa and the theoretical isoelectric point of the protein was 9.12 according to ExPASy-ProtParam. The FvMYB114 protein was composed of 99 aa, among which Leu, Arg, Glu and Ala residues accounted for 12.1%, 10.1%, 9.1% and 8.1%, respectively. In addition, the FvMYB114 protein was hydrophilic as its overall mean hydrophilic coefficient was −0.743.

Sequence alignment of the FvMYB114 protein with MYB proteins from nine other species revealed that FvMYB114, like these proteins, included a conserved DNA-binding domain of MYB, which was typical of the 1R-MYB TF family (Figure 1A). Figure 1B shows the results of phylogenetic tree analysis. It can be seen that FvMYB114 and *Rosa chinensis* RcMYB14 (UNZ22460.1) had the highest homology and the closest genetic relationship. Analysis of the secondary structure of FvMYB114 protein revealed that it contained 45.45% α-helix, 12.12% β-coil, 4.04% extended strand and 38.38% random coil (Figure 2A). As shown in Figure 2B, at 8–58 aa, the aa sequence of FvMYB114 contained a conserved SANT domain. All the results indicated that FvMYB114 belonged to the 1R-MYB family. In addition, the tertiary structure of the FvMYB114 protein was predicted using the SWISS-MODEL website, and it was found that the FvMYB114 protein had a HTH region (Figure 2C), which was consistent with the predicted secondary structure.

### 2.2. FvMYB114 Was Localizated to Nucleus

35S::FvMYB114-GFP construction was transferred to tobacco epidermal cells using the *Agrobacterium tumefaciens* infiltration method. The results of confocal microscope observation are shown in Figure 2. The fluorescence of 35S::GFP construct as a control can be observed in the whole cell (Figure 3A), but the fluorescence signal of 35S::FvMYB114-GFP fusion protein can only be observed in the nucleus (Figure 3E). Therefore, it could be preliminarily determined that FvMYB114 protein was located in the nucleus.

### 2.3. Expression Level Analysis of FvMYB114 in F. vesca Seedlings

The qRT-PCR results for *FvMYB114* in *F. vesca* organs (young leaves, mature leaves, stems and roots) under a control condition are shown in Figure 4A, indicating that *FvMYB114* had a high expression level in young leaves and roots, but a low expression level in stems and mature leaves (Figure 4A). The expression level in young leaves was 1.5, 2.7 and 3.2 times higher than that in roots, stems and mature leaves, respectively. Furthermore, in young leaves, the expression of *FvMYB114* under five stresses (cold, salt, dehydration, heat and ABA) showed a trend of rising first and then falling; the peak expression times were 5 h, 7 h, 9 h, 3 h and 7 h, respectively, and then expression began to decline. In roots, gene expression also showed a similar trend, with peak times of expression at 7 h, 5 h, 5 h, 7 h and 3 h, respectively. In addition, in these two organs, the highest expression level of *FvMYB114* under low temperature and salt stress was higher than that under other treatments (Figure 4B,C). These results indicated that *FvMYB114* was more sensitive to cold and salt than other stress stimuli.

### 2.4. Overexpression of FvMYB114 in A. thaliana Enhanced Salt Tolerance

In order to explore the role of *FvMYB114* in high salt and low temperature environments, transgenic *Arabidopsis* were placed under these two conditions. WT and UL lines were used as controls. Through qPCR detection, it was found that among 12 transgenic *A. thaliana* plants, only five gene lines with a high expression level (L1, L2, L3, L4, L5) were obtained with the addition of *FvMYB114*-specific primers, among which the expression of *FvMYB114* in L1, L2 and L4 was high, as shown in Figure 5A.

All the plants with good growth (WT, UL, L1, L2, L4) were irrigated with 200 mM NaCl. After one week of treatment, the phenotypes of each plant were observed. It can be seen that the leaves of the WT and UL lines were severely chlorotic, the plants were small and there was an obvious phenomenon of shrinkage and wilting. However, for transgenic lines, only parts of the plants showed marginal yellowing. After irrigation with water for three days, WT and UL withered seriously, but the transgenic lines recovered and gradually resumed their growth (Figure 5B). The survival rates of WT, UL, L1, L2 and L4 were 22%, 26%, 73%, 79% and 75%, respectively (Figure 5C). It can be concluded that overexpression of *FvMYB114* significantly improved the survival rate of transgenic *Arabidopsis* under salt stress.

The contents of O_2_^−^ and H_2_O_2_ were measured (Appendix A), and under a control condition, no differences were observed for any lines. Salt treatment increased the content of O_2_^−^/H_2_O_2_ in WT, UL, L1, L2 and L4. The O_2_^−^ appeared significantly different between WT and *FvMYB114* transgenic *Arabidopsis.* The O_2_^−^ contents of WT, UL, L1, L2 and L4 increased by 2.36, 2.20, 1.41, 1.29 and 1.39 times more compared to the control, while the H_2_O_2_ contents of WT, UL, L1, L2 and L4 increased by 2.11, 2.30, 1.56, 1.65 and 1.58 times more than the control. This showed that under salt stress, a large amount of ROS was accumulated in plant cells, which would cause damage to plants. However, due to the overexpression of *FvMYB114*, ROS accumulation in plant cells was lower. It can be seen from Figure 6 that under the control conditions, the measured physiological and biochemical indicators did not significantly differ among the plants. However, after seven days of growth under high-salt conditions, the CAT, SOD and POD activities and chlorophyll and proline contents of *FvMYB114* transgenic *Arabidopsis* were higher than those of the WT and UL lines. For MDA, its content was higher in WT and UL. In triplicate trials, after salt stress treatment, compared with WT lines under the control condition, the chlorophyll contents of transgenic lines decreased by 0.58, 0.62 and 0.53 times, the MDA contents increased by 0.06, 0.06 and 0.38 times, the proline contents increased by 3.4, 4.8 and 3.42 times, CAT activity increased by 1.21, 1.02 and 1.11 times, SOD activity increased by 2.45, 1.62 and 2.1 times and POD activity increased by 1.37, 1.13 and 1.27 times. These results indicate that overexpression of *FvMYB114* reduced the degree of membrane lipid peroxidation of transgenic *Arabidopsis* when faced with salt stress, and also strengthened plants’ ability to remove ROS, so they had greater viability under salt stress.

### 2.5. Expression Analysis of Salt-Resistant Genes in FvMYB114-OE A. thaliana

Regulation of target gene expression at the transcriptional level is a key link in the plant stress response. In order to clarify the potential mechanism of stronger salt tolerance in *FvMYB114* overexpression lines, the expression levels of four proven *A. thaliana* response genes (*AtSOS1*, *AtSOS3*, *AtNHX1* and *AtLEA3*) [27] to salt stress in all plants (WT, UL, L1, L2, L4) were analyzed. The results are shown in Figure 7. Under the control condition, the expression of these four genes was largely equal. After seven days of salt stress treatment, their expression levels were significantly increased, but it can be clearly seen that the expression levels of functional genes in L1, L2 and L4 were much higher than in the WT and UL. From these data, we can infer that *FvMYB114* can enable plants to better cope with salt stress by upregulating the expression of *AtSOS1*, *AtSOS3*, *AtNHX1* and *AtLEA3*.

### 2.6. Overexpression of FvMYB114 Improved the Cold Tolerance of A. thaliana

As shown in Figure 8, when *A. thaliana* (WT, UL, L1, L2, L4) was in a normal environment, the phenotypes were almost and the plants were in a healthy state and growing well. However, when the growth environment was changed and the temperature was reduced to −8 °C, after 14 h of growth under such conditions, the phenotypes of these plants had significant differences. The number and area of leaves of WT and UL plants decreased, while the damage to L1, L2 and L4 was not serious. Then, these plants were returned to room temperature, and after one week of recovery, it was found that the growth of the WT and UL could not be restored due to the severe impact of low temperature, with survival rates of only 22% and 26%, respectively. Compared with that, the survival rates of transgenic lines L1, L2 and L4 were higher at 76%, 77% and 80%, respectively (Figure 8B).

The results in Figure 9 show that under the control condition (22 °C), the contents of chlorophyll, MDA and proline and the activities of POD, SOD and CAT of each line were largely equal, with little difference. However, after low-temperature stress, the activity or content of other indicators increased (except for the chlorophyll content, which decreased). The activities of POD, SOD, CAT, chlorophyll and proline of transgenic *A. thaliana* were much higher than those of the WT and UL, while the content of MDA was lesser than those of the WT and UL. In triplicate trials, after cold stress treatment, compared with the WT lines under the control condition, the chlorophyll contents of transgenic lines decreased by 0.15, 0.4 and 0.2 times, the MDA contents increased by 0.05, 0.25 and 0.05 times, the proline contents increased by 3.98, 4.15 and 4.05 times, CAT activity increased by 1.36, 1.07 and 1.14 times, SOD activity increased by 2, 1.82 and 1.98 times and POD activity increased by 2.17, 2.35 and 0.96 times. Furthermore, the O_2_^−^ and H_2_O_2_ contents were calculated (Appendix A), and we found no differences between the lines under the control condition, while the contents of the WT and UL increased more sharply than those of L1, L2 and L4 with the cold treatment. The O_2_^−^ contents of the WT, UL, L1, L2 and L4 increased by 2.50, 2.25, 1.44, 1.39 and 1.43 times compared to the control, and the H_2_O_2_ contents of the WT, UL, L1, L2 and L4 increased by 2.22, 2.13, 1.60, 1.52 and 1.58 times compared to the control. That is to say, high expression of *FvMYB114* is helpful to enhance the tolerance of plants to low temperature.

### 2.7. Expression Analysis of Cold-Resistant Genes in FvMYB114-OE A. thaliana

In order to further explore the regulatory mechanism of *FvMYB114* in response to plant low-temperature stress, in this study, we analyzed the changes in the expression levels of four cold response functional genes—*AtCBF1*, *AtCBF3*, *AtNHX1* and *AtLEA3*—in *A. thaliana*. It can be seen from Figure 10 that under low temperature conditions, the expression of these four genes increased, indicating that they can respond to cold, but the increase differed. The expression amount in overexpressed lines was significantly higher than that in the WT and UL lines. Therefore, overexpression of *FvMYB114* can improve the expression of cold response genes.

## 3. Discussion

The growth and development of plants are easily affected by the changing environment. As sessile organisms, plants have evolved complex defense systems to cope with various environmental stresses [28,29]. As the largest family of TFs in plants, MYB can participate in the response of plants to abiotic stress in various regards, including affecting plant metabolism, participating in metabolic reactions and regulating hormone signal transduction, to enable plants to carry out normal growth and development and ensure that plant yield will not be reduced [30,31]. It is of great significance for crop improvement and breeding to understand the mechanism of MYB TF in plants responding to complex environments.

In this study, WT forest strawberry was used as the experimental material, and specific primers were designed using the DNAMAN software with homologous cloning technology, with the *FvMYB114* gene sequence obtained by cloning. Sequence analysis results showed that *FvMYB114* contained a 300 bp nucleotide sequence, which encoded 99 aa (Figure 1). This 300 bp sequence is one (Fv2339_5g36520-mRNA-2_Fv2339_v1.0) of the three possible transcripts of the *FvMYB114* gene located on chromosome 5 (https://www.rosaceae.org/ (accessed on 25 December 2022) GDR (rosaceae.org)). *FvMYB114* contained a unique SANT-MYB DNA-binding conservative domain, which is typical of a 1R-MYB gene (Figure 2). Subcellular localization showed that FvMYB114 was a nuclear localization protein (Figure 3). According to the phylogenetic tree results, FvMYB114 and RcMYB114 had the closest genetic relationship, which indicated that *FvMYB114* can participate in regulating plant responses to abiotic stressors, as *RcMYB114* does [32]. The gene expression pattern can provide important clues that help explore the gene function. Therefore, in order to verify our conjecture, the expression level of the target gene was analyzed.

During plant development, gene products are expressed in different tissues or organs in different orders, that is, gene expression is cell- or tissue-specific. Figure 4 showed that the expression of *FvMYB114* in *F. vesca* conformed to this characteristic. In the same period, the expression of *FvMYB114* was the lowest in mature leaves and the highest in young leaves, followed by roots. This may be because *FvMYB114* was most sensitive to stress signals in active organs. In roots and immature leaves, different stresses could induce the expression of *FvMYB114*, and its expression level changed with time. In this experiment, *FvMYB114* may have been sensitive to cold and salt stress, which provided a basis for further analysis. Salt and cold stress will hinder the growth and development of plants and limit the productivity of crops [33,34]. It has been found that MYB TF plays an important role in the process of plant response to stress. In this study, WT, UL and transgenic lines were placed in a high-salt and low-temperature environment (Figure 5 and Figure 8). After a period of treatment, although all *A. thaliana* were damaged to some extent, the WT and UL lines suffered more seriously than the transgenic lines. This indicated that overexpression of *FvMYB114* could enhance the tolerance of *A. thaliana* to high salt and cold, which was consistent with the results of previous studies.

The morphological characteristics and physiological and biochemical changes of plants in a stress environment are an expression of their resistance to stress [35]. When plants are under stress, they will rapidly produce a large number of ROS (ROS broadly referring to free radicals and non-free radicals from oxygen sources), which throw the redox state in cells out of balance and cause serious damage to cells [36]. In order to detect how the content of ROS in the cells of plants changed under stress, we detected the contents of O_2_^−^ and H_2_O_2_. It can be seen from the test results that *Arabidopsis* accumulated a large amount of ROS in cells under stress, but due to the overexpression of *FvMYB114*, ROS accumulation in plant cells was lower. The protective enzyme system mainly includes SOD, POD and CAT, which can remove ROS from cells in a harsh growth environment to reduce damage to cells. Therefore, their activities can be used to reflect the damage degree of plants under an adverse environment [37]. As the final product of membrane lipid peroxidation, MDA can be used to indicate the degree of membrane lipid peroxidation and reflect plant damage [38]. The metabolic balance of chlorophyll will be broken by pressure, thus changing the content of chlorophyll. Therefore, it can be used to indicate the strength of plant resistance [39]. Proline, as a component of plant protein, can regulate the osmotic balance between the cytoplasm and the vacuole and enhance the tolerance of plants to stress [40,41]. In this study, after salt and cold treatment, CAT, SOD and POD activities of all lines increased, and these activities increased significantly in *FvMYB114* transgenic *Arabidopsis* (Figure 6 and Figure 9). Meanwhile, the MDA and chlorophyll contents decreased less and the proline content increased more in transgenic plants than in the UL and WT. These results indicated that high-salt and low-temperature stress could induce *FvMYB114* to start the response mechanism, and the overexpression of *FvMYB114* had a significant role in improving plant stress resistance.

ABA plays an important role in the plant salt stress response, and MYB TF is involved in the ABA signal transduction pathway [42,43,44]. The biosynthesis of ABA and the signal sent by ABI1 can enhance the expression of *NHX1* at the transcription level, and the *NHX* gene family can use the H^+^ gradient to transport Na^+^ to the vacuole for storage, thus reducing the toxicity of Na^+^ [45]. *NHX1* has the potential to improve the tolerance of plants to salt stress, and overexpression of the *SbNHX1* gene can increase the tolerance of *Salicornia brachiata* to NaCl stress [46]. LEA protein has great significance for improving plant salt tolerance. It has been found that the overexpression of *OsLEA3* in rice could positively regulate the response of ABA to salt stress and drought stress [47]. The SOS signal transduction pathway is the key way to maintain intracellular ion homeostasis and improve salt stress tolerance. SOS1, SOS2 and SOS3 are very helpful to improve plant salt tolerance in this way [48,49]. In our study, the expression levels of four target genes related to salt stress in the *FvMYB114*-OE line were upregulated, giving plants salt tolerance (Figure 7). Therefore, we speculate that *FvMYB114* may be able to regulate the response of plants to salt stress through an ABA-dependent pathway and the SOS signal transduction pathway.

In the process of evolution, plants have formed a response mechanism to cold stress, which is mainly controlled by the CBF pathway, supplemented by other pathways [50]. Studies have shown that MYB TF can bind to the CBF promoter region to increase the cold-related gene expression; meanwhile, CBF can also improve the expression of CRT/DRE cis-acting elements of genes such as RDs, CORs and LTls to promote plants’ cold resistance. Overexpression of *MdMYB23* in apple or *Arabidopsis* can promote the expression of *MdCBF*s, thereby enhancing the cold tolerance of plants [51]. In this study, we analyzed the expression changes of four genes of the CBF transduction pathway in *Arabidopsis* after low-temperature treatment (Figure 10). It was found that the expression of these four genes was high in transgenic lines, which was consistent with the previous research results [17,25,26]. Therefore, it is speculated that *FvMYB114* can regulate the expression of cold response genes through the CBF pathway, to improve plant cold tolerance.

To sum up, we have speculated on the molecular mechanism of *FvMYB114* in response to low-temperature stress and salt stress based on the experimental results and compiled a possible functional model (Figure 11). When subjected to cold stimulation, *FvMYB114* overexpressed to upregulate the expression of cold response genes *CBF1* and *CBF3* of the CBF pathway; *CBF1* and *CBF3* also improved the expression of CRT/DRE cis-acting elements of genes *AtCOR4* and *AtCCA1* to further enhance the cold resistance of plants. When receiving the salt stress signal, *FvMYB114* increased its expression after being stressed and regulated the response of plants to salt stress in two ways: one was to promote the expression of *SOS1* and *SOS3* through the SOS pathway, and the other was to improve the expression level of *NHX1* and *LEA3* genes through the ABA signal transduction pathway.

It has also been proven in other species that the expression of these stress-related genes is low in the absence of stress. In previous studies, it was found that the *CBF* gene was transferred into *V. riparia*, and under cold treatment, the expression of the *CBF* gene could increase over a period of time [52]. After salt stress treatment, the expression of *AtNHX1* in kiwi fruit increased, improving the salt tolerance of kiwi fruit [53]. Calzone et al. [54] identified and characterized the expression of *NHX1* and *SOS1* genes, which play a key role in salt tolerance. Only when there was stress was their expression significantly increased. As our results showed (Figure 7 and Figure 10), the expression of transgenic *Arabidopsis* was higher than that of the WT in the absence of stress, but not significantly higher. We infer that when these functional genes encounter environmental stress, in transgenic *Arabidopsis*, *FvMYB114* works with other factors to regulate the co-expression of genes. Accordingly, the expression level of transgenic *Arabidopsis* under stress was significantly higher than that of the WT. We speculate that in addition to *FvMYB114*, there are other TFs or other factors involved in regulating the response of plants to environmental factors. In general, the salt tolerance mechanism of *FvMYB114* needs to be further explored.

## 4. Materials and Methods

### 4.1. Plant Materials, Growth Conditions and Treatment

The *F. vesca* seedlings from Harbin China were seeded on Murashige and Skoog (MS) media containing indole-3-butyric acid (IBA) (Solarbio, A8170, Beijing, China) and 6-benzylaminopurine (6BA)(Solarbio, A8170, China) (both plant hormone concentrations were 0.6 mg/L) or in a substrate with a ratio of soil to vermiculite of 2:1. They were placed in an incubator (Jinghong GZP-350Y, Shanghai, China)with a temperature of 25 °C and a relative humidity of 70%, and a photoperiod of 16 h/8 h (light/dark) was set [55]. When 9 true leaves of the seedlings showed, the unfolded young leaves, completely expanded mature leaves, newly appeared roots and newly appeared stems were selected in 10 *F. vesca* seedlings to test the relative mRNA levels. Then, we selected 30 seedlings with good growth and divided them into 6 groups with 5 seedlings in each group. One group was placed in the tissue culture room at 25 °C as the control, and 2 groups were placed in the incubator at 4 °C and 37 °C, respectively, for low temperature and high temperature [56]. The other 3 groups were treated with 15% PEG6000 (Solarbio, P8250, Beijing, China), 200 mM NaCl (Solarbio, LA0200, Beijing, China) and 100 uM ABA (Solarbio, A8060, Beijing, China), respectively, to simulate drought, salt and ABA stress. To determine the cold treatment results, the roots and young leaves of all 6 groups’ seedlings were sampled at 0, 1, 3, 5, 7, 9 and 11 h, when they were immediately frozen with liquid nitrogen and then stored at −80 °C for subsequent RNA extraction [57].

### 4.2. Isolation and Cloning of FvMYB114

Base on the transcriptome database of Strawberry Genomic Resources (http://bioinformatics.towson.edu/strawberry/ (accessed on 14 March 2022); gene ID: gene26513) and the transcriptome database of our lab (gene ID: Fvb2-2-gene-17.54), cold-related gene *FvMYB114* of *F. vesca* was screened and isolated. Total RNA was extracted from roots, stems and leaves (immature and mature leaves) using an OminiPlant RNA Kit (Conway Collection, Beijing, China) and purified using RNase-Free DNase I. The first-strand cDNA was then synthesized using TransScript First Strand cDNA Synthesis SuperMix (TransGen Biotech, Beijing, China). We selected a pair of gene-specific primers (*FvMYB114*-F and *FvMYB114*-R; Appendix A). The cDNA was used as a template to amplify the target fragment via PCR, and then the PCR product was ligated with PEASY-T1 vector (TransGen Biotech, Beijing, China) to screen out positive colonies and send them to sequencing [58].

### 4.3. Subcellular Localization of FvMYB114

Primers of *BamHI* and *SalI*, which are two enzyme digestion sites contained in the polyclonal site region of pSAT6-GFP-N1 vector (*FvMYB114*-slF and *FvMYB114*-slR; Appendix A), were designed. Then, the vector and target protein FvMYB114 were double-digested with *SalI* and *BamHI* restriction endonucleases. Following that, the target fragment of *FvMYB114* was inserted between *SalI* and *BamHI* to obtain the transient expression vector of FvMYB114 [59]. The fusion plasmid containing FvMYB114 and the empty 35S::GFP plasmid as a control were injected into the outer tobacco epidermal cells usig the *A. tumefaciens* injection method for subcellular localization. The location of the FvMYB114-GFP fusion protein was examined using confocal microscopy (LSM 510 Meta, Zeiss, Germany).

### 4.4. Sequence Analysis and Structure Prediction of FvMYB114

The multiple sequences of *FvMYB114* and MYB TF of other species were compared using DNAMAN 5.2. We utilized MEGA7 to construct a phylogenetic tree via the neighbor-joining method (http://www.megasoftware.net (accessed on 15 November 2021)) [60]. We then predicted the primary structure of the FvMYB114 protein using the ExPASy website (ProtParam tool: https://web.expasy.org/protparam/ (accessed on 23 November 2021)). The domain of the FvMYB114 protein was predicted uing the SMART website (http://smart.embl-heidelberg.de/ (accessed on 23 November 2021)), and the tertiary structure of FvMY114 protein was predicted on the SWISS-MODEL website (https://swissmodel.expasy.org/ (accessed on 23 November 2021)).

### 4.5. Expression Analysis of FvMYB114

We used qPCR to detect the expression level of *FvMYB114* in different tissues under a control condition and abiotic stress. According to the conservative series of *FvMYB114*, we designed qPCR primers *FvMYB114*-qF and *FvMYB114*-qR (Appendix A). The reaction system was as follows: 2xMix 12.5 μL, ddH2O 9 μL, 1.5 μL cDNA, 1 μL primer. The reaction procedure was: 94 °C for 30 s, 95 °C for 5 s, 54 °C for 40 s, 72 °C for 30 s; perform 40 cycles at 72 °C for 10 min; and store the PCR product at 4 °C [61]. The internal reference gene was the *Actin* gene (XM_011471474.1, *F. vesca*), and its primers are shown in Appendix A (*FvActin*-F, *FvActin*-R). The expression levels of relative mRNA in roots, stems, young leaves and mature leaves under the control condition were statistically analyzed using one-way ANOVA with Tukey’s test. The expression levels of the target gene in young leaves and roots under abiotic stress were analyzed using the 2^−∆∆*C*t^ method [62].

### 4.6. Obtaining Transgenic A. thaliana

The PCAMBIA2300 vector was digested with *BamHI* and *SalI* restriction enzymes. Then, the target fragment of *FvMYB114* was inserted into *BamHI* and *SalI* of the PCAMBIA2300 vector to construct the pCAMBIA2300-*FvMYB114* overexpression vector. The vector pCAMBIA2300-*FvMYB114* was transferred into *A. tumefaciens* GV3101, and GV3101-mediated *A. tumefaciens* was transferred into *Arabidopsis* Columbia via the inflorescence-mediated method. Next, *FvMYB114*-OE *Arabidopsis* was generated by the CaMV35S promoter. MS medium containing 50 mg/L kanamycin was used to screen transgenic lines. After qPCR analysis, the transgenic lines were finally determined, and the WT and UL were set as the control group. Follow-up analysis was conducted with T_3_ generation transgenic lines. The relative expression level of *FvMYB114* in transgenic *A. thaliana* was statistically analyzed by using one-way ANOVA with Tukey’s test.

### 4.7. Stress Treatment and Determination of Related Physiological Indexes in A. thaliana

*A. thaliana* of WT, UL and T_3_ transgenic lines (L1, L2, L4) were planted in 1/2 MS medium. After 10 days, the cotyledons were exposed, and the seedlings were transferred to nutrient pots (soil:vermiculite = 2:1). Four seedlings were planted in each pot. All *A. thaliana* were divided into 3 groups for different stress treatments (each group contained 20 plants). One group served as a control and grew under normal conditions, while the other two groups were treated with the following treatments. One group was placed in an incubator at −8 °C for 14 h and then returned to room temperature for growth for 7 days. The other group was irrigated with 200 mM sodium chloride for 7 days under salt stress and then irrigated with water for 3 days. After treatment, the phenotypic characteristics of each plant were observed and the survival rate of each line was calculated. The samples used for the determination of physiological and biochemical indicators were all WT, UL and transgenic lines after stress treatment. We determined the absorbance of the extracted chlorophyll solution according to the method put forward by Wang [63], and calculated the content of chlorophyll according to the formula proposed by Wellburn [64]. The activities of CAT, SOD and POD were measured according to the methods of Kirham et al. [65]. The MDA content was determined with an MDA Analysis Kit (TBA method) [66]. We measured and calculated the proline contents in the samples as set out by Qu et al. [67]. Finally, we measured the contents of O_2_^−^ and H_2_O_2_ according to the method outlined by Espinosa et al. [68].

### 4.8. Analysis of Genes Expression of FvMYB114

The mRNA of the WT, UL and L 1/2/4 lines grown under a normal condition, salt stress or low-temperature stress were extracted and reverse transcribed into the first-strand cDNA, which was used as template, while *AtActin* was used as the internal reference. The functional genes of *Arabidopsis thaliana* to MYB were tested using the qRT-PCR process and system, as described in Section 4.5.

### 4.9. Statistical Analysis

SPSS 21.0 software (IBM, Chicago, IL, USA) was used to analyze the one-way variance. All data were the means of triplicate trials, and the standard deviation (SD) was calculated. Statistical differences were referred to as significance. * *p* ≤ 0.05, ** *p* ≤ 0.01.

## 5. Conclusions

In this study, we isolated and cloned *FvMYB114* with transcriptional activity from *F. vesca* and characterized the expression of *FvMYB114* as tissue-specific and high in young leaves and roots. After functional analysis, it was found that the overexpression of *FvMYB114* played a key role in enhancing the response of plants to stress. From the current results, it can be concluded that *FvMYB114* can improve the tolerance of plants to salt stress and low temperature.

## Figures and Tables

**Figure 1 ijms-24-05261-f001:**
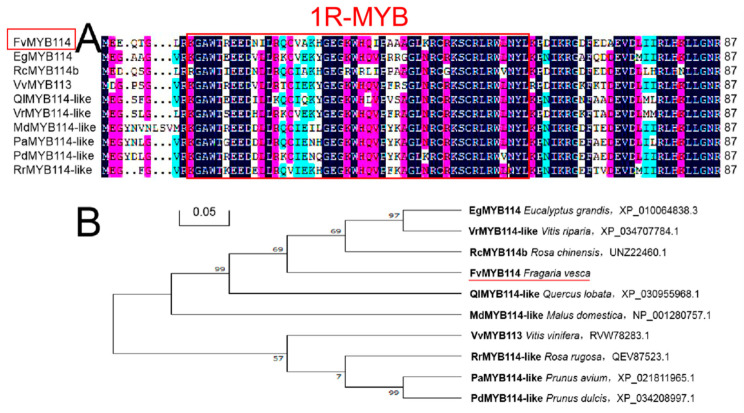
Comparison between FvMYB114 and MYB proteins of other species and their homologous evolution. (**A**) Analysis of homology between FvMYB114 and MYB proteins of other species. The red box is the conservative region of the amino acid sequence, which is typical for 1R-MYB TF. (**B**) Phylogenetic tree analysis of FvMYB114 and nine other MYB proteins, where FvMYB114 is marked with a red line. The accession numbers are as follows: EgMYB114 (XP_010064838.3), RcMYB114b (UNZ22460.1), VvMYB113 (RVW78283.1), QlMYB114-like (XP_030955968.1), VrMYB114-like (XP_034707784.1), MdMYB114-like (NP_001280757.1), PaMYB114-like (XP_021811965.1), PdMYB114-like (XP_034208997.1) and RrMYB114-like (QEV87523.1). The red underline is the target protein.

**Figure 2 ijms-24-05261-f002:**
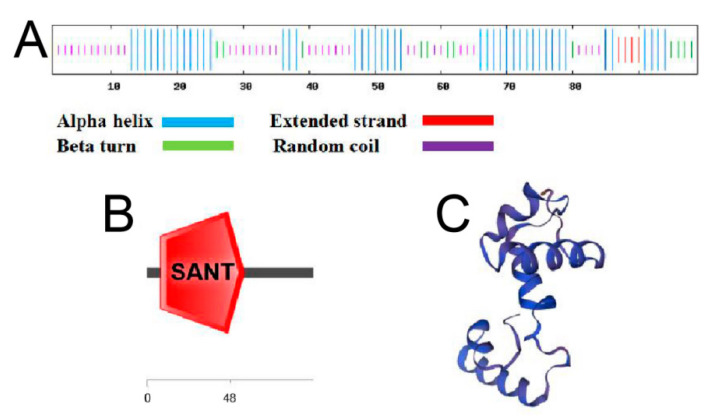
Structure and domains of predicted FvMYB114 protein. (**A**) Secondary structure; (**B**) protein domains; (**C**) tertiary structure of FvMYB114 protein.

**Figure 3 ijms-24-05261-f003:**
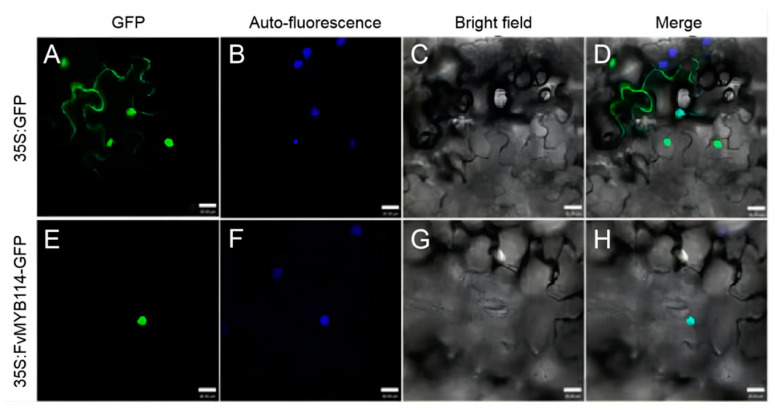
Subcellular localization of FvMYB114. The 35S:GFP and 35S:FvMYB114 plasmids were transformed into the cells by particle bombardment. (**A**,**E**) GFP fluorescence; (**B**,**F**) Auto-fluorescence; (**C**,**G**) Bright-field images; (**D**,**H**) Merged. Bar = 50 μm.

**Figure 4 ijms-24-05261-f004:**
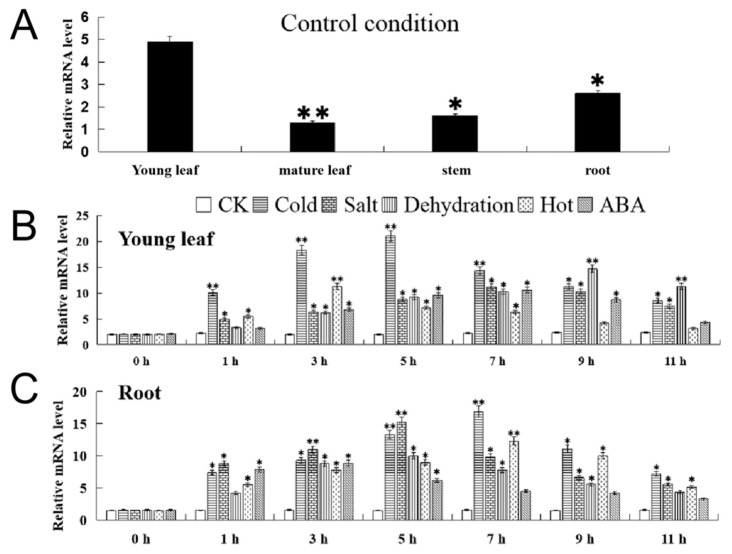
Real-time expression of *FvMYB114* gene. (**A**) The expression level of *FvMYB114* in young leaves, mature leaves, stems and roots. (**B**,**C**) Expression of *FvMYB114* in young leaves and roots after different times and treatments. The standard deviation is indicated by an error line. An asterisk indicates that there is a significant difference (SD) between the treatment group and the control group, * *p* ≤ 0.05, ** *p* ≤ 0.01, Student’s *t*-test.

**Figure 5 ijms-24-05261-f005:**
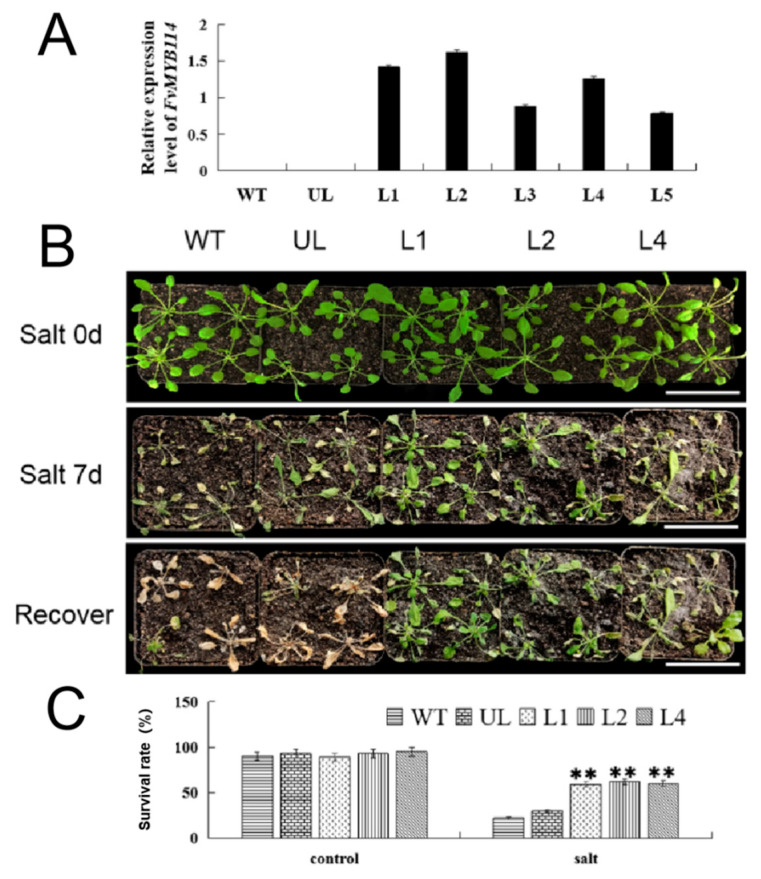
Growth of *FvMYB114*-OE *A. thaliana* after salt stress. (**A**) The relative expression levels of *FvMYB114* in WT, UL and *FvMYB114*-OE lines. (**B**) The phenotypes of WT, UL and transgenic lines (L1, L2 and L4) under control conditions, salt stress and stress recovery conditions. Bar = 5 cm. (**C**) Survival rates of each line (WT, UL, L1, L2 and L4) under different treatments. ** SD between transgenic lines and WT and UL, *p* ≤ 0.01.

**Figure 6 ijms-24-05261-f006:**
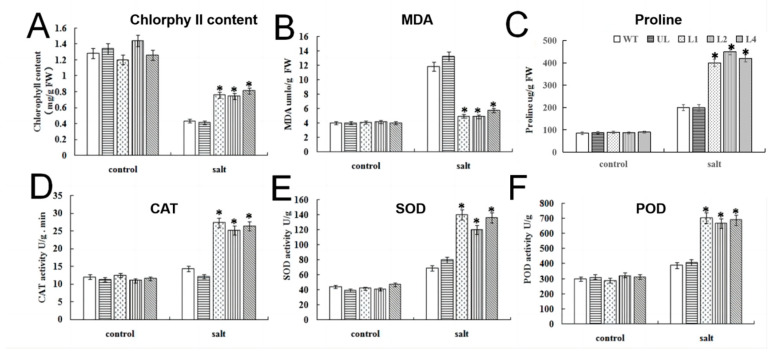
Effects of salt stress on physiological and biochemical indexes of *FvMYB114* transgenic lines. After being treated with 200 mM NaCl for 7 days, the contents of (**A**) chlorophyll, (**B**) MDA and (**C**) proline and the activities of (**D**) CAT, (**E**) SOD and (**F**) POD in WT, UL, L1, L2 and L4 were determined. An asterisk marks the SD between transgenic lines and WT (* *p* ≤ 0.05). The control group set the level of each index in the WT.

**Figure 7 ijms-24-05261-f007:**
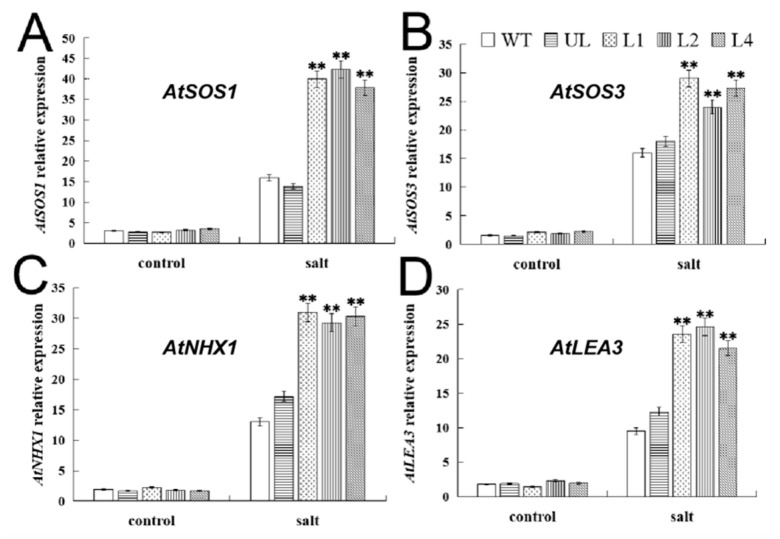
The expression levels of genes related to salt stress in each strain. Relative expression levels of (**A**) *AtSOS1*, (**B**) *AtSOS3*, (**C**) *AtNHX1* and (**D**) *AtLEA3*. SD is represented by error bars. The data are the average of the three tests. ** SD with WT, *p* ≤ 0.01.

**Figure 8 ijms-24-05261-f008:**
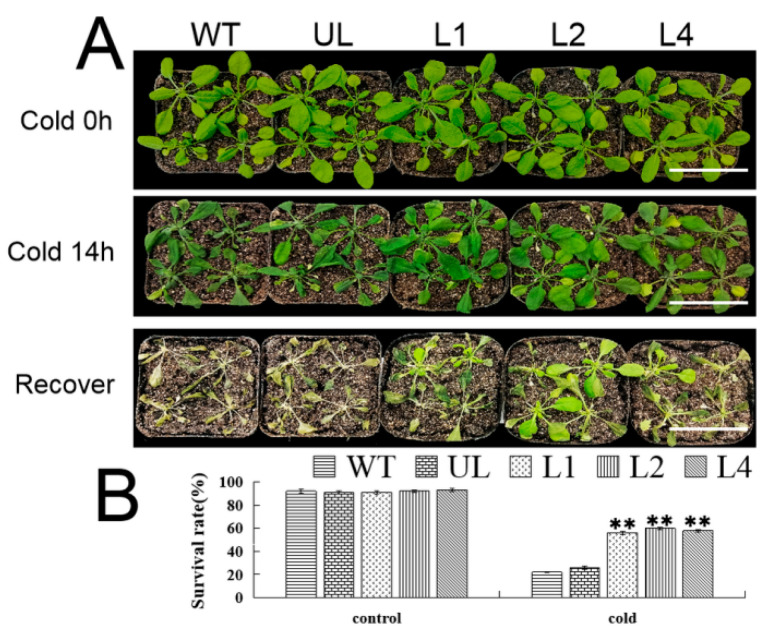
Growth of *FvMYB114*-OE *A. thaliana* after salt stress. (**A**) The phenotypes of WT, UL and transgenic lines (L1, L2 and L4) under control conditions, cold stress and stress recovery conditions. Bar = 5 cm. (**B**) Survival rates of each line (WT, UL, L1, L2 and L4) under different treatments. ** SD between transgenic lines and WT and UL, *p* ≤ 0.01.

**Figure 9 ijms-24-05261-f009:**
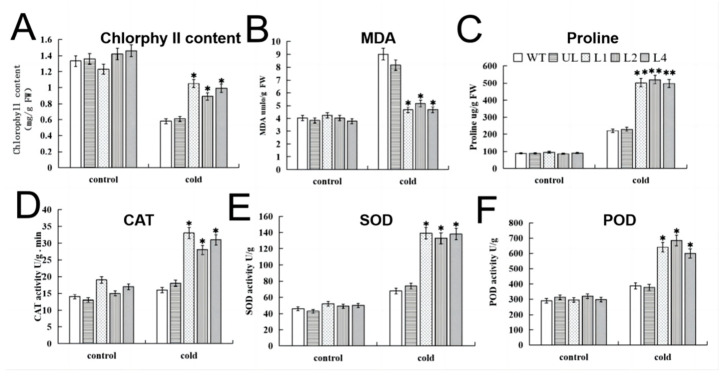
Effects of cold stress on physiological and biochemical indexes of *FvMYB114* transgenic lines. After being treated with cold, the contents of (**A**) chlorophyll, (**B**) MDA and (**C**) proline and the activities of (**D**) CAT, (**E**) SOD and (**F**) POD in WT, UL, L1, L2 and L4 were determined. An asterisk marks the SD between transgenic lines and WT (* *p* ≤ 0.05, ** *p* ≤ 0.01). The control group set the level of each index in WT.

**Figure 10 ijms-24-05261-f010:**
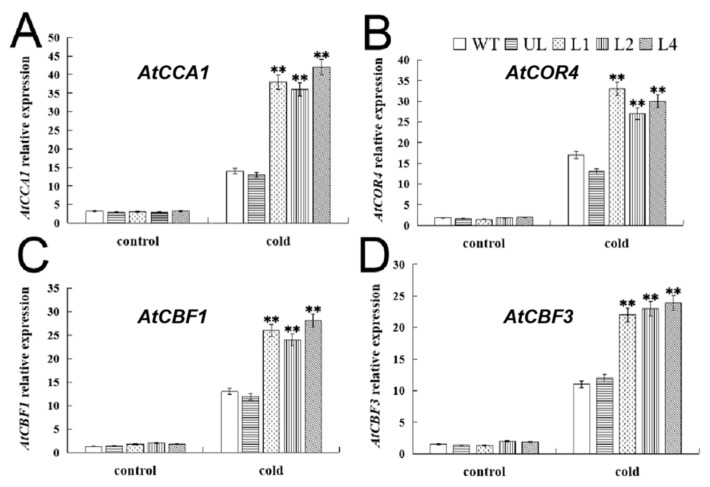
The expression levels of genes related to cold stress in each strain. Relative expression levels of (**A**) *AtCCA1*, (**B**) *AtCOR4*, (**C**) *AtCBF1* and (**D**) *AtCBF3*. SD is represented by error bars. The data are the averages of three tests. ** SD with WT, *p* ≤ 0.01.

**Figure 11 ijms-24-05261-f011:**
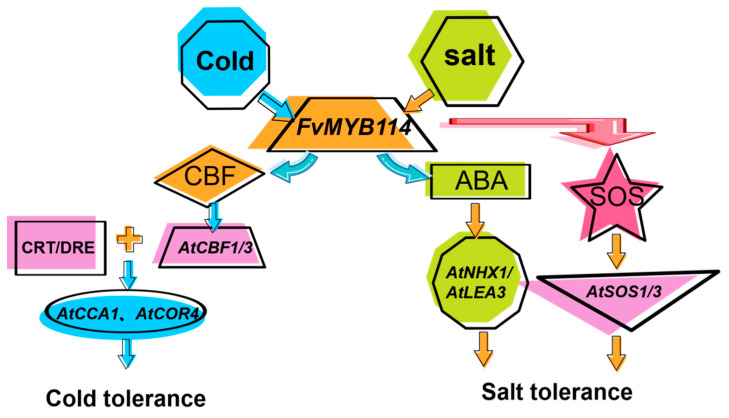
A possible functional model of *FvMYB114* in response to low-temperature stress and salt stress. When subjected to cold stimulation, *FvMYB114* upregulates the expression of cold response genes *CBF1*, *CBF3*, *COR4* and *CCA1* located through the CBF pathway. FvMYB114 binds to the promoter region of *CBF1/3*, thereby promoting the binding of CBFs to the CRT/DRE cis-acting elements of genes, finally activating the expression of cold-related genes *CCA1* and *COR4.* When receiving a salt stress signal, *FvMYB114* can promote the expression of *SOS1* and *SOS3* through the SOS pathway and can also improve the expression levels of *NHX1* and *LEA3* genes through the ABA signal transduction pathway.

## Data Availability

Not applicable.

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
