# Peer review of "Overexpression of a Fragaria vesca 1R-MYB Transcription Factor Gene (FvMYB114) Increases Salt and Cold Tolerance in Arabidopsis thaliana"

_ijms, 2023, doi:10.3390/ijms24065261_

Round 1

Reviewer 1 Report (New Reviewer)

Li et al., manuscript focuses on the identification and ectopic overexpression of  FvMYB114. After careful consideration, I find the manuscript is not in publishable form. I hope the following comments would help authors in reshaping the MS  for future submissions.

> The main issue of the study is that the authors mention the length of the gene is 300 bp. When I looked at the currently available genome assemblies of the Fragaria vesca, the potential candidate genes are not matching the said size. It was further verified after checking the primer sequences provided by the authors.  

> Also, methods are not elaborated adequately in order to reproduce the study by others. As an example Figure 4 A, there is no mention of how the relative mRNA levels were measured (what condition is the normalizer?).  

> Please correct errors related to scientific nomenclature including indicating the authority of species.

> Please recheck the manuscript and correct all errors related to gene nomenclature.

> There are numerous spelling, formatting, and grammar mistakes. I highly recommend the authors obtain the assistance of an academic editing service.

Other comments and suggestions are given in the attached file.

Author Response

Li et al., manuscript focuses on the identification and ectopic overexpression of  FvMYB114. After careful consideration, I find the manuscript is not in publishable form. I hope the following comments would help authors in reshaping the MS for future submissions.

> The main issue of the study is that the authors mention the length of the gene is 300 bp. When I looked at the currently available genome assemblies of the Fragaria vesca, the potential candidate genes are not matching the said size. It was further verified after checking the primer sequences provided by the authors.  

Thank you for your suggestion, we added the length of this gene in supplementary files (figure S1).

> Also, methods are not elaborated adequately in order to reproduce the study by others. As an example Figure 4 A, there is no mention of how the relative mRNA levels were measured (what condition is the normalizer?).  

Thank you for your suggestion,we correct it. The normalizer is the mature leaf.

> Please correct errors related to scientific nomenclature including indicating the authority of species.

Thank you for your suggestion, we correct it.

> Please recheck the manuscript and correct all errors related to gene nomenclature.

Thank you for your suggestion, we correct it.

> There are numerous spelling, formatting, and grammar mistakes. I highly recommend the authors obtain the assistance of an academic editing service.

Thank you for your suggestion

Other comments and suggestions are given in the attached file.

Reviewer 2 Report (New Reviewer)

In this manuscript, a 1R-MYB TF gene was obtained from Fragaria vesca,and overexpression of FvMYB114 enhanced the tolerance of Arabidopsis thaliana to salt and low temperature. Here, I have some comments that authors should address and consider for the revision.

1.    It has been reported that MYB transcription factors participate in plant salt tolerance and cold resistance in Arabidopsis, the authors speculate on molecular mechanism of FvMYB114 by overexpression in Arabidopsis, the research methods lack of innovation, it will be better to study the function of FvMYB114 in Fragaria vesca.

2.    Expression level analysis of FvMYB114 showed that the expression of FvMYB114 in roots was significantly lower than that in leaves, why do they show a similar expression trend under different stresses?

3.    The authors indicated that FvWRKY114 was more sensitive to cold and salt than other stress stimuli from figure 4, I think this conclusion maybe not rigorous, under other stresses, the expression of FvWRKY114 was also higher than that of the control. In addition, the authors did not perform phenotypic experiments under other stresses.

4.    The columns of figure 4B and 4C showed that the expression level of FvMYB114 under other stresses was significantly higher than that of the control (such as hot in 4B at 7h, ABA in 4C at 5h…), but the authors indicated that there were no significant differences, I suggest the authors reanalyze the data.

5.    What is the statistical number of survival rate in the figure 5C and figure 8B? It is better for the author to provide the number of plants in the paper.

6.    Overexpression of FvMYB114 greatly promoted the expression of genes related to salt stress and cold stress, whether the promoters of these genes have common specific elements that can be bound by FvMYB114? What is the downstream target gene directly bound by FvMYB114? Which target gene is dominant in response to salt stress and cold stress?

7.    The ordinate in the figure 5C is not survival rate.

8.    Tobacco cells in figure 3 are not clear, the author should provide clearer pictures.

9.    The font size in figures and format of references should be standardized and unified.

10.  I recommend that you have your paper professionally edited for English language.

Author Response

In this manuscript, a 1R-MYB TF gene was obtained from Fragaria vesca,and overexpression of FvMYB114 enhanced the tolerance of Arabidopsis thaliana to salt and low temperature. Here, I have some comments that authors should address and consider for the revision.

  1. It has been reported that MYB transcription factors participate in plant salt tolerance and cold resistance in Arabidopsis, the authors speculate on molecular mechanism of FvMYB114 by overexpression in Arabidopsis, the research methods lack of innovation, it will be better to study the function of FvMYB114 in Fragaria vesca.

Thank you for your suggestion, the former MYB transcription factor we identified responding to salt and cold tolerance was R2R3-MYB (FvMYB82),this is the first 1R-MYB TF we found participated in salt and cold stress, meanwhile we will study the function of FvMYB114 in Fragaria vesca next step.

  1. Expression level analysis of FvMYB114 showed that the expression of FvMYB114 in roots was significantly lower than that in leaves, why do they show a similar expression trend under different stresses?

Under different stresses, roots and leaves showed similar expression trends, but the expression levels in different parts were different, and the expression peak time points were also different. For example, under cold stress, the expression level reached the peak at 5h in young leaves was 23, and reached the peak at 7h in roots was 17, indicating that young leaves responded to cold stress more quickly and strongly than roots under cold stress.

  1. The authors indicated that FvMYB114 was more sensitive to cold and salt than other stress stimuli from figure 4, I think this conclusion maybe not rigorous, under other stresses, the expression of FvMYB114 was also higher than that of the control. In addition, the authors did not perform phenotypic experiments under other stresses.

Thank you for your suggestion, it is true under other stresses, the expression of FvMYB114 was also higher than that of the control, but under cold and salt condition the relative mRNA level was highest, that is the reason why we focused on cold and salt stress. And the phenotypic experiments showed no significant differences between stress and control within 11h, but with qPCR the relative mRNA level could show the differences.

  1. The columns of figure 4B and 4C showed that the expression level of FvMYB114 under other stresses was significantly higher than that of the control (such as hot in 4B at 7h, ABA in 4C at 5h…), but the authors indicated that there were no significant differences, I suggest the authors reanalyze the data.

Thank you for your suggestion, we correct it.

  1. What is the statistical number of survival rate in the figure 5C and figure 8B? It is better for the author to provide the number of plants in the paper.

Thank you for your suggestion, we correct it.

  1. Overexpression of FvMYB114 greatly promoted the expression of genes related to salt stress and cold stress, whether the promoters of these genes have common specific elements that can be bound by FvMYB114? What is the downstream target gene directly bound by FvMYB114? Which target gene is dominant in response to salt stress and cold stress?

Thank you for your suggestion, in this paper we had not identify which target gene is dominant in response to salt and cold stress. Next step we will find these genes promoter and with Yeast one-hybrid experiment to ensure the directly downstream target gene.

  1. The ordinate in the figure 5C is not survival rate.

Thank you for your suggestion, we correct it.

     8. Tobacco cells in figure 3 are not clear, the author should provide clearer pictures.

Thank you for your suggestion, we correct it.

    9. The font size in figures and format of references should be standardized and unified.

Thank you for your suggestion, we correct it.

    10. I recommend that you have your paper professionally edited for English language.

Thank you for your suggestion, we correct it.

Reviewer 3 Report (New Reviewer)

I reviewed the paper titled: Overexpression of a Fragaria vesca MYB transcription factor gene (FvMYB114) increases salt and cold tolerance in Arabidopsis thaliana

The authors obtained a new 1R-MYB TF gene from Fragaria vesca (a diploid strawberry) and given a new name, FvMYB114 and showed that overexpression of FvMYB114 greatly enhanced the adaptability and tolerance of Arabidopsis thaliana to salt and low temperature. The results are very interesting and important for strawberry industry. I only have a few concerns that needs to be addressed by the authors as follow:

1-      How many copies of the gene was inserted to your transgenic lines. Did you do southern blot? If not how can you answer the biosafety concerns in this regards?

2-      As I see you had 5 transgenic lines. Are 5 lines enough for releasing a transgenic variety?

3-      Is the amount of expression of this gene in your transgenic lines is enough for releasing the variety?

4-      I suggest to do a better literature review about drought tolerance in fruits and the genes involved in drought tolerance and cite them in introduction and discussion. For example, the following papers can be addressed:

http://jast.modares.ac.ir/article-23-5369-en.html

https://doi.org/10.1080/14620316.2020.1812446

https://doi.org/10.1016/j.scienta.2018.05.024

Author Response

I reviewed the paper titled: Overexpression of a Fragaria vesca MYB transcription factor gene (FvMYB114) increases salt and cold tolerance in Arabidopsis thaliana

The authors obtained a new 1R-MYB TF gene from Fragaria vesca (a diploid strawberry) and given a new name, FvMYB114 and showed that overexpression of FvMYB114 greatly enhanced the adaptability and tolerance of Arabidopsis thaliana to salt and low temperature. The results are very interesting and important for strawberry industry. I only have a few concerns that needs to be addressed by the authors as follow:

  • How many copies of the gene was inserted to your transgenic lines. Did you do southern blot? If not how can you answer the biosafety concerns in this regards?

Single copy. In the 12 transgenic lines tested, 5 strains were screened with single copy of FvMYB114 gene. We did not do southern blot, but qPCR was done for biosafety concerns in this regards.

  • As I see you had 5 transgenic lines. Are 5 lines enough for releasing a transgenic variety?

Thank you for your advice. More than 5 transgenic lines we got,as other person reported before 5 lines are enough for releasing a transgenic variety.

  • Is the amount of expression of this gene in your transgenic lines is enough for releasing the variety?

Yes. The expression amount of FvMYB114 in transgenic lines is high enough for releasing the variety.

  • I suggest to do a better literature review about drought tolerance in fruits and the genes involved in drought tolerance and cite them in introduction and discussion. For example, the following papers can be addressed:

Thank you for your suggestion, we had addressed the following papers in introduction.

http://jast.modares.ac.ir/article-23-5369-en.html

https://doi.org/10.1080/14620316.2020.1812446

https://doi.org/10.1016/j.scienta.2018.05.024

Round 2

Reviewer 1 Report (New Reviewer)

Before moving forward, please provide the original chromatograms of the FvMYB114 received from the sequence provider for the review process.

Author Response

Reviewer 2 Report (New Reviewer)

This revised manuscript thas been modified as recommended.

Author Response

This revised manuscriptthas been modified as recommended.

Round 3

Reviewer 1 Report (New Reviewer)

Dear Authors,

Thank you for the response. However, I still find that you worked with a partial transcript. I attached the blast results of a newer genome assembly version FYI compared to the SGR: Strawberry Genomic Resources.

Author Response

This manuscript is a resubmission of an earlier submission. The following is a list of the peer review reports and author responses from that submission.

Round 1

Reviewer 1 Report

1. The sentence "In addition, WRKY protein can combine W-box [TGACC (A/T)] in its target  gene promoter." in the abstract is irrelevant to the full text. It is suggested to delete this sentence.

2. Figure 3. image of subcellular localization is not clear enough. It is recommended to re-experiment

3. Figure 4A "yong leaf "should be amended to" young leaf "

4. The stress treatment time in fig. 4 was inconsistent with that in method 4.1

5. Dehydration stress in Figure 4 is not mentioned in the method.

6. Fig. 5B Phenotype is not obvious after 7 days of salt stress treatment.

7. The callout number is marked incorrectly since 2.7

8. Figure 8A "cold 14h "should be amended to" cold 10h "

9. Error in title, 2.6 Expression Analysis of Cold Resistant Downstream Genes in FvMYB114 -OE A. thaliana should be 2.8….

10. The discussion is too lengthy, just retelling the results.

Reviewer 2 Report

A new MYB TF gene was cloned from Fragaria vesca (a diploid strawberry) in a manuscript entitled Overexpression of a Fragaria vesca MYB transcription factor gene (FvMYB114) increases salt and cold tolerance in Arabidopsis thaliana. MYB genes are a part of a large family of transcription factors and play important roles participate in plant development and responses to stresses. The authors make a systematic contribution to the research literature in this area of investigation. However, Certain changes are needed to improve the overall quality of the manuscript. The manuscript could be considered for publication with the following suggestion and corrections.

1.    Please modify the font size [email protected].

2.    Please modify and rephrase the abstract and remove such statements (After the introduction of FvMYB114) and (After salt and cold stress treatment)

3.    In the Abstract, how do you know WRKY protein can combine W-box? Please explain

4.    Why did you choose Fragaria vesca for the isolation of MYB TF gene?

5.    MYB TFs also participate in drought. Why did the authors only focus on low temperature and salt stress?

6.    Result part 2.1. There is a statement (the protein was predicted to be hydrophilic).  what can be the purpose of this nature please explain.

7.    Two subheadings have been 2.3, be consistent with subheadings numbering.

8.    Pay attention to the abbreviations and significant terms throughout the manuscript.

9.    Results part 2.3 Overexpression of FvMYB114 in A. thaliana enhanced Salt tolerance, what VvWRKY114 stands for?

10. I have found a few typographical errors, please pay attention to spelling throughout the manuscript.

11.  Please make sure the statistical analysis is correct

12. The conclusion needs to be consistent with scientific terms.

13. All figures should be the same font size and can be further improved.

14. The language of the manuscript needs further improvement.

Author Response

  1. Please modify the font size [email protected].

Response: Yes, We have accepted your suggestion and have modified the font size [email protected].

  1. Please modify and rephrase the abstract and remove such statements (After the introduction of FvMYB114) and (After salt and cold stress treatment)

Response: Yes, We have accepted your suggestion and have modified and rephrased the abstract.

  1. In the Abstract, how do you know WRKY protein can combine W-box? Please explain

Response: We know it based on previous studies and relevant articles, but we think that this sentence is irrelevant in the full text, so we have deleted it.

  1. Why did you choose Fragaria vesca for the isolation of MYB TF gene?

Response: MYB TF has a variety of functions, but it mainly focuses on the regulation of volatile substances, anthocyanins and flavonol accumulation in strawberry fruit. There are few reports on the function of MYB gene in Fragaria vesca salt tolerance and cold resistance. And The genetic transformation system of F. vesca is stable, it can be used as the dominant subgenome of cultivated strawberry.

  1. MYB TFs also participate in drought. Why did the authors only focus on low temperature and salt stress?

Response: In our experiment, we found that FvMYB114 was more sensitive to salt and cold stress than drought stress. Therefore, we mainly studied the effect of FvMYB114 on plant response to low temperature and salt stress.

  1. Result part 2.1. There is a statement (the protein was predicted to be hydrophilic).  what can be the purpose of this nature please explain.

Response: We are to better understand the physical and chemical properties of this protein.

  1. Two subheadings have been 2.3, be consistent with subheadings numbering.

Response: Yes, We have accepted your suggestion and used the correct order.

  1. Pay attention to the abbreviations and significant terms throughout the manuscript.

Response: Yes, We have accepted your suggestion and checked and revised abbreviations and important terms throughout the manuscript.

  1. Results part 2.3 Overexpression of FvMYB114 in A. thaliana enhanced Salt tolerance, what VvWRKY114 stands for?

Response: It should be FvMYB114. Due to our negligence, we wrote the gene name incorrectly.

  1. I have found a few typographical errors, please pay attention to spelling throughout the manuscript.

Response: Yes, We have accepted your suggestion and checked and revised spelling throughout the manuscript.

  1. Please make sure the statistical analysis is correct

Response: Yes, we have made sure the statistical analysis is correct

12.The conclusion needs to be consistent with scientific terms.

Response: Yes, we have accepted your suggestion and have modified this part.

  1. All figures should be the same font size and can be further improved.

Response: Yes, we have accepted your suggestion and adjusted the font size of all numbers to the same

  1. The language of the manuscript needs further improvement.

Response: Yes, we have accepted your suggestion and improved the language of the manuscript.

Reviewer 3 Report

The authors studied the effect of F. vescea MYB transcription factor (FvMYB114) on salt and cold stress tolerance in transgenic Arabidopsis. Transgenic plants showed higher activities of enzymatic (SOD, CAT, POD) and non-enzymatic (proline) antioxidants, and a lower level of MDA. The authors assumed that the FvMYB114 improved stress tolerance by promoting the expression of the genes AtSOS1/3, AtNHX1, AtLEA3 (salt stress) and the genes AtCCA1, AtCOR4, AtCBF1/3  (cold stress). The data obtained are interesting and worth publishing. However, some questions need to be more discussed.

Considering that MYB genes play a role in the defence response during plant stress, then one would expect that an overexpression of FvMYB114 would affect the expression of the above genes even in non-stressed transgenic Arabidopsis, which was not the case (Figure 6, 7). What changed if the studied genes were upregulated under stress conditions, if FvMYB114 expression in transgenic plants was constitutive? It seems that there are other factors as well, so Figure 11 may not fully reflect reality.  This should be discussed.

Others:

The Figures should be uniform in size and easily visible for reading (eg Fig. 3 and Fig. 4)

Although Agrobacterium-mediated transformation is now a common method, it should be mentioned in Materials and Methods

Figs. 4, 5a, 6, 9 - Did the authors really use  2−ΔΔCt method?

Statistical significance in individual Figures should more readable, to distinguish between transgenic lines and WT. At the very least, it would be good to make all statistical comparisons to the unstressed WT plants.

There is no information on how many plants/line  were statistically analyzed.

Round 2

Reviewer 1 Report

Authors have revised my concerns

Author Response

The comments of reviewers have been revised.

Reviewer 3 Report

The authors did not answer my questions sufficiently and there are still inconsistencies that they have not been able to resolve.

For example:

Still unclear evaluation of qPCR:  the authors used the delta delta cT method once (M and M 4.5) and then probably the delta cT method (M and M 4.8).  A bit unusual.  

There is still not a clear how many plants/line/stress were analyzed (e.g. qPCR, MDA, chlorophylls...). The information they provide is still confusing.

The statistical evaluation of individual experiments is confusing, it would be appropriate to use another type of test, not only the T-Test

Even though the authors supplemented M and M with a part of Arabidopsis transformation, it is still not clear which promotor was used to drive the studied gene. Plasmid CAM3011 – there is no citation.

The authors did not answer my questions sufficiently and there are still inconsistencies that they have not been able to resolve.

For example:

Still unclear evaluation of qPCR:  the authors used the delta delta cT method once (M and M 4.5) and then probably the delta cT method (M and M 4.8).  A bit unusual.  

There is still not a clear how many plants/line/stress were analyzed (e.g. qPCR, MDA, chlorophylls...). The information they provide is still confusing.

The statistical evaluation of individual experiments is confusing, it would be appropriate to use another type of test, not only the T-Test

Even though the authors supplemented M and M with a part of Arabidopsis transformation, it is still not clear which promotor was used to drive the studied gene. Plasmid CAM3011 – there is no citation.

The authors have still not answered this question:

„Considering that MYB genes play a role in the defence response during plant stress, then one would expect that an overexpression of FvMYB114 would affect the expression of the above genes even in non-stressed transgenic Arabidopsis, which was not the case (Figure 6, 7). What did change if the studied genes were upregulated under stress conditions, if FvMYB114 expression in transgenic plants was constitutive? It seems that there are other factors as well, so Figure 11 may not fully reflect reality.  This should be discussed. “

Author Response

The authors did not answer my questions sufficiently and there are still inconsistencies that they have not been able to resolve.

For example:

Still unclear evaluation of qPCR:  the authors used the delta delta cT method once (M and M 4.5) and then probably the delta cT method (M and M 4.8).  A bit unusual.  

There is still not a clear how many plants/line/stress were analyzed (e.g. qPCR, MDA, chlorophylls...). The information they provide is still confusing.

The statistical evaluation of individual experiments is confusing, it would be appropriate to use another type of test, not only the T-Test

Response: Thank you for your suggestion. This time we used T-test. In the next test, we will consider using ANOVA and other analysis methods.

Even though the authors supplemented M and M with a part of Arabidopsis transformation, it is still not clear which promotor was used to drive the studied gene. Plasmid CAM3011 – there is no citation.

Response: The CaMV35S promoter drives the studied gene, this part has been supplemented in materials and methods. It should be pCAMBIA2300, because of our mistakes, CAM3011 was written.

The authors have still not answered this question:

„Considering that MYB genes play a role in the defence response during plant stress, then one would expect that an overexpression of FvMYB114 would affect the expression of the above genes even in non-stressed transgenic Arabidopsis, which was not the case (Figure 6, 7). 

Response: Fig. 6 and 7 under no environmental stress, the expression amount of downstream genes of FvMYB114-OE Arabidopsis overexpression was increased compared with that of WT, but the expression amount was not significant, because all downstream genes were stress related genes, and the expression amount was significantly increased only under stress, and the expression amount in transgenic Arabidopsis was more significant than that in wild type.

What did change if the studied genes were upregulated under stress conditions, if FvMYB114 expression in transgenic plants was constitutive? It seems that there are other factors as well, so Figure 11 may not fully reflect reality.  This should be discussed. “

Response: Thank you for your thoughtful comments. We have added this part to the discussion section, and we will also study the influence of other factors in future experiments, so that we can understand more thoroughly the role of the target gene in plant stress response.

Round 3

Reviewer 3 Report

The authors still did not sufficiently respond to my comments.